# Sound in Sights: Deriving Visual Insights from Audio for Comprehensive Video Understanding with Large Multimodal Models

## Abstract

Video understanding is inherently multimodal, requiring both visual and auditory cues to form a complete representation of dynamic scenes. However, most existing video understanding models rely solely on visual content, overlooking informative audio cues, such as spoken instructions or environmental sounds, for scene understanding and event comprehension. Progress in audio-visual reasoning has been hindered by the lack of high-quality supervised fine-tuning (SFT) data that jointly considers video and audio. To address this gap, we introduce SoundInSights , a large-scale audio-visual question answering dataset comprising over $80k$ question–answer pairs from online videos, created via a multimodal large language model (MLLM)-assisted annotation pipeline. SoundInSights provides rich supervision for audio-visual reasoning, enabling MLLMs to be fine-tuned for complex joint audio-video understanding. We find that current video MLLMs heavily rely on visual information, hindering effective multimodal learning. To mitigate this, we propose an audio-only pretraining stage, which significantly improves audio-visual reasoning performance. Additionally, to evaluate audio-visual comprehension, we construct a high-quality, manually curated test set of 1,000 samples requiring joint audio-visual understanding, exceeding standard benchmarks in complexity. Models fine-tuned on SoundInSights with the proposed training strategy achieve substantial performance gains on this new benchmark. Moreover, on the challenging VideoMME evaluation, our approach significantly improves performance in Information Synopsis subcategory, demonstrating the efficacy of incorporating audio. The SoundInSights dataset and code will be publicly released to facilitate further research.

## 1 Introduction

Video multimodal large language models (MLLMs) aim to extend large language models (LLMs) with the ability to perceive and comprehend complex visual content and temporal relationships in videos. Despite significant advancements, most state-of-the-art models Zhang et al. (2024); Wang et al. (2024a); Shen et al. (2024); Li et al. (2023b); Sun et al. (2024); Maaz et al. (2023); Chen et al. (2024b); Li et al. (2024); Ye et al. (2024a); Chen et al. (2024c); Wang et al. (2024b) focus exclusively on visual inputs, entirely neglecting auditory signals. This oversight is critical, as real-world videos inherently contain rich auditory cues essential for accurate interpretation. Many important events occur off-screen, where sound alone—such as a honking car or approaching footsteps—can reveal crucial contextual information. Recent research highlights these limitations, demonstrating scenarios where certain actions are visually "invisible" yet acoustically informative, or visually distinct yet acoustically indistinguishable Huh et al. (2023). Furthermore, conversational scenarios rely heavily on audio cues like tone of voice and ambient noises, which visual-only models miss. Ignoring audio thus not only restricts model perception but diverges significantly from how humans naturally integrate visual and auditory information. Bridging this gap in multimodal perception demands models that *listen* as effectively as they *see*. A few approaches have attempted to address this audio perception gap. Some works utilize audio captioning modules to convert auditory input into textual captions, subsequently feeding these into LLMs Fu et al. (2024); Jeoung et al. (2024). However, this text-centric approach inherently compresses and limits the richness of audio representations and

Figure 1: **Reasoning Invisible Event**: SoundInSights model integrates audio and video modalities, enabling effective reasoning about events that are invisible.

prevents direct audio-visual interaction. Other approaches Zhang et al. (2023); Cheng et al. (2024); Shu et al. (2023); Sun et al. (2024); Wang et al. (2024c) attempt to jointly embed audio and video into a unified multimodal space, but these efforts have been hindered by the scarcity of high-quality joint audio-video datasets necessary for training robust cross-modal representations.

The emergence of large multimodal language models presents a promising opportunity to advance video understanding significantly. MLLMs, such as LLaVA and LLaVA-Video, demonstrate the powerful capability of generating rich, contextually informative annotations, thereby enabling efficient scaling of high-quality synthetic datasets. For instance, LLaVA-Video showcases how multimodal annotations for videos can be produced efficiently at scale by leveraging GPT-4-driven question-answer pair generation. While existing datasets like *WavCaps* Mei et al. (2024) and *AudioCaps* Kim et al. (2019) have made strides in providing annotations for audio or video individually, they typically lack sophisticated, structured annotations that fully exploit multimodal context.

To fully unlock the potential of MLLMs in video understanding, models must learn to seamlessly fuse auditory and visual modalities, dynamically interpreting temporal relationships and contextual nuances. Current MLLM-based frameworks largely treat modalities independently or through simplified fusion techniques, falling short of capturing the intricate interplay between sight and sound. Developing truly multimodal MLLMs requires sophisticated training strategies, richer multimodal datasets, and explicitly designed architectures to handle complex audio-visual reasoning tasks.

In this paper, we introduce a *novel multimodal video dataset*, **SoundInSights dataset**, specifically constructed to bridge this significant gap in audio-visual comprehension. Central to our approach is a *hierarchical audio annotation schema* designed to offer structured, multi-level insights into the auditory dimension of videos. Instead of treating audio as a singular descriptive caption, we annotate audio at multiple granularities—from detailed short-interval captions to broader contextual summaries—thus fostering nuanced audio comprehension. Complementing these annotations, we also generate high-quality, context-dependent audio-visual QA pairs using MLLMs. Our QA design ensures questions require integrated cross-modal reasoning, explicitly testing models' ability to synthesize information across sight and sound.

To facilitate scalability, we leverage a highly efficient automated annotation pipeline driven by multimodal large language models, notably ChatGPT. This pipeline first systematically annotates audio at multiple granularities and subsequently generates context-aware QA pairs reflecting deep audio-visual integration. Compared to traditional annotation approaches, our method significantly enhances both annotation efficiency and complexity, enabling the processing of extensive video content rapidly and comprehensively.

We believe our work provides foundational advances that can significantly enhance the capabilities and research trajectories in multimodal video understanding. By highlighting the necessity of deeply integrating auditory signals into video comprehension frameworks, our contributions pave the way toward more sophisticated, robust, and human-like multimodal reasoning systems.

## 2 RELATED WORK

### 2.1 AUDIO-VISUAL DATA CURATION

Early multimodal datasets primarily addressed audio-visual integration through specialized or constrained contexts. For example, the AVSD dataset Alamri et al. (2019) provides human-dialogue-based

QA annotations for approximately 11k short videos but remains limited due to its narrow domain (Charades actions) and high annotation costs. Similarly, VGG-Sound Chen et al. (2020) provides extensive coverage with 200k videos spanning 309 predefined sound categories, yet its reliance on closed-set classification limits flexibility, precluding open-ended questions essential for broader multimodal understanding.

Recently developed benchmarks have started exploring open-ended audio-visual question answering, though often within restricted domains. The MUSIC-AVQA dataset Li et al. (2022) extends question diversity yet remains constrained to musical performance scenarios. The AVQA dataset Yang et al. (2022), while providing audio-visual questions, predominantly employs template-based questions that limit reasoning complexity to basic recognition tasks. The AVinstruct dataset Ye et al. (2024b) further attempts to enrich annotations by rephrasing existing questions with concise video descriptions, yet does not substantially broaden context richness or complexity.

In summary, prior datasets exhibit limitations including restricted question diversity, constrained domains, and limited scale, leaving substantial room for datasets featuring diverse, open-ended questions and scalable multimodal annotations that cover realistic, everyday audio-visual scenarios.

### 2.2 Multimodal Large Language Model for Video Understanding

The rise of multimodal LLMs has spurred methods to integrate visual and auditory information for video understanding. Approaches like X-InstructBLIP Panagopoulou et al. (2023) and OneLLM Han et al. (2024) extend instruction-tuned LLMs to handle not just images but also audio and video inputs, aiming for a unified cross-modal encoder. Others such as Video-LLaMA Zhang et al. (2023) (and its successor VideoLLaMA2 Cheng et al. (2024)) incorporate audio tracks alongside video frames via pretrained encoders and alignment modules. Models including PandaGPT Su et al. (2023), Macaw-LLM Lyu et al. (2023), AV-LLM Chowdhury et al. (2025), and AVicuna Tang et al. (2024) have been trained on available audio-visual datasets to enhance LLM understanding of videos. However, most of these systems still lean heavily on visual information, treating audio as an auxiliary feature. In practice, their audio-visual integration remains shallow – e.g. some use encoders trained mostly on image data – and truly joint reasoning over sound and vision is underdeveloped. Indeed, current multimodal models tend to underemphasize the role of audio in audio-visual reasoning tasks. This underscores that effective audio-visual reasoning capability in LLMs is still in its infancy, often constrained by the limitations of the training data and strategies used.

### 2.3 Supervised Fine-Tuning for Audio-Visual Learning

Supervised fine-tuning of multimodal models on joint video-audio tasks has so far been bottlenecked by the available training data. Many works resort to repurposing or synthesizing datasets that are not originally designed for complex AV reasoning. For example, the VALOR dataset Chen et al. (2023) provides audio-visual captions but only for trimmed 10-second clips, while QA-oriented sets like AVSD Alamri et al. (2019) or MUSIC-AVQA Li et al. (2022) are small-scale or domain-specific (and lack fine-grained temporal annotations). To compensate, recent multimodal LLM efforts have constructed massive instruction-tuning corpora by combining or generating QA pairs. VideoLLaMA2, for instance, uses ∼100k GPT-generated video instruction–response pairs for tuning, and models like OneLLM and AVicuna were fine-tuned on roughly 460k and 350k audio-visual Q&A examples respectively. Such brute-force data augmentation highlights the scarcity of high-quality, diverse AV training data. In contrast, our proposed dataset offers a substantially larger and richer set of directly annotated audio-visual QA pairs, avoiding the need for purely synthetic data. By covering a broad spectrum of realistic AV scenarios with varied question types, it provides a more effective fine-tuning resource for multimodal LLMs. Our accompanying training pipeline leverages this data to instill stronger audio-visual reasoning capabilities, leading to models that significantly outperform those trained on previous datasets in understanding and answering questions about complex audio-visual content.

## 3 SoundInSights

High-quality multimodal datasets are essential for training effective multimodal large language models (MLLMs)Zhang et al. (2024). However, existing video datasets face two primary challenges:

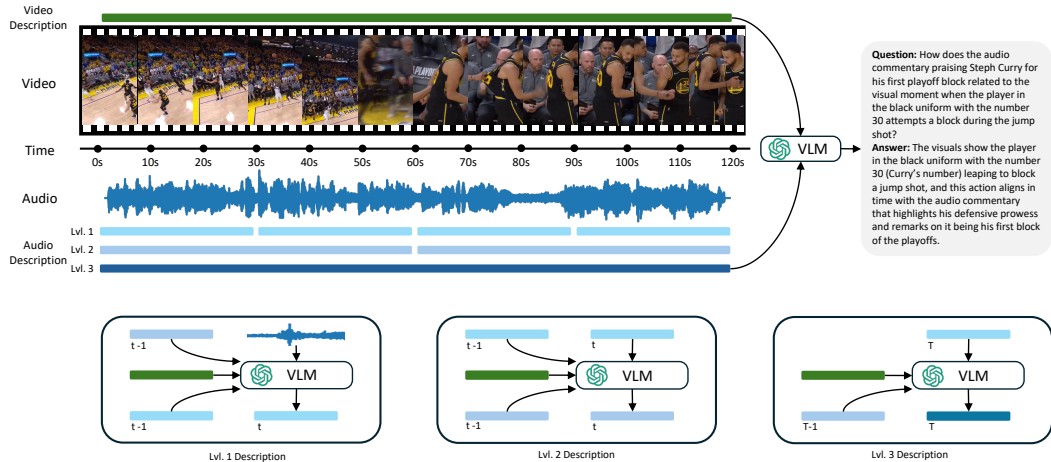

Figure 2: Pipeline of generating hierarchical audio descriptions and audio-video joint QA pairs.

(1) sparse audio annotations that fail to comprehensively capture nuanced sound semantics, and (2) insufficiently detailed correlations between audio and visual modalities, limiting robust multimodal reasoning. Although the importance of richer multimodal datasets is widely recognizedGong et al. (2024), progress has been hampered by manual labeling's prohibitive costs and complexities involved in modality alignment.

Inspired by recent advances in automated data annotation and synthesis methods Zhang et al. (2024); Chen et al. (2024a), we can utilize some modality-specific MLLMs Hurst et al. (2024); Team et al. (2024) to provide multi-modal information in text format, and then leverage LLMs to extract the information from their results to generate high-quality conversation dataset.

Leveraging their capabilities, we propose a scalable framework designed to automatically generate high-quality conversation data. Comparing with previous audio-video datasets like AVInstruct Ye et al. (2024b) and AVQA Yang et al. (2022), our SoundInSights dataset has the following features: 1) detailed auditory annotations that incorporate visual context for richer semantic understanding; 2) Complex open-ended question answering pairs which requires a joint understanding of the visual contents and the audio in the input videos.

**Data Source.** To ensure diversity and effectiveness, our annotation pipeline leverages videos sourced from Youtube videos of LLaVA-Video 178K Zhang et al. (2024), enabling comprehensive auditory and visual instructional learning.

## 3.1 HIERARCHICAL AUDIO ANNOTATIONS

Generating precise and temporally aligned audio descriptions is essential for creating natural and detailed audio-visual conversational data. However, previous studies often overlook this critical aspect. For instance, AVInstruct employs subtitle generators (e.g., BLIP2 Li et al. (2023a), Whisper Radford et al. (2023)) to produce global audio descriptions, which lack detail and struggle with effectively handling long-duration videos.

To address this issue, our audio annotation methodology employs GPT-4o Audio Hurst et al. (2024), a state-of-the-art audio understanding model, as the primary annotation tool. Additionally, inspired by the hierarchical annotation strategy proposed in LLaVA-Video-178k Zhang et al. (2024), we implement a three-level hierarchical annotation structure, carefully balancing annotation granularity with contextual coherence. Each hierarchical level uniquely contributes to enhancing multimodal reasoning capabilities.

Specifically, videos are segmented into shorter clips that are individually fed into the audio multimodal large language model (MLLM). Concurrently, each audio segment is paired with its corresponding visual description. This visual context is crucial for resolving inherent ambiguities present in audio

alone, such as distinguishing subtle audio events like a *beep* from a *strike*. Furthermore, the visual context helps to explicitly link audio elements with their corresponding visible objects.

As illustrated in Figure 2, we generate audio captions at three distinct hierarchical levels, with detailed inputs as follows:

**Level-1 (Clip-Level).** For every 30-second interval, detailed annotations describe the audio events by integrating: the current *audio clip*, the most recent segment summary (Level-2), and the entire video's visual narrative.

**Level-2 (Segment-Level).** At 1-minute intervals, we summarize cumulative audio information to capture broader contextual coherence. Inputs include: the recent Level-1 annotation, the previous summary, and overall visual context.

**Level-3 (Global-Level).** A comprehensive final synopsis is produced, encapsulating the entire audio content by integrating: remaining recent Level-1 annotations, and the most recent level-2 annotation.

This hierarchical structure enhances annotation quality, enabling more precise multimodal reasoning by clearly delineating temporal and contextual audio events. The exact algorithm is provided in Appendix A.

## 3.2 Audio-Visual Question Answering

To further facilitate multimodal reasoning, we automatically generate challenging QA pairs explicitly requiring integrated audio-visual understanding. This automated pipeline involves three key steps:

**Data Compilation.** We gather annotated audio clips, their hierarchical summaries, and detailed visual descriptions, establishing a robust multimodal context.

**Question Formulation.** Questions are designed to prioritize audio-centric information, demanding reasoning that synthesizes auditory details with visual context, while explicitly avoiding references to raw annotations.

**QA Generation and Filtering.** Using advanced GPT models (e.g., GPT-o1 mini Jaech et al. (2024)), we produce and refine QA pairs, applying stringent validation to ensure clarity, relevance, and multimodal dependency. This ensures the generated QAs necessitate integrated reasoning, effectively challenging models' multimodal capabilities.

---

**Sample Data**

**Clip Caption:**
1. The clip begins with an enthusiastic greeting, "What's up everybody?...
2. The clip starts with the speaker confidently stating...
...

**Audio Summary:**
The clip begins with an enthusiastic and friendly greeting... As background remains quiet, the speaker goes on to describe the illusion, posing a challenge with the question, "What do you see?"... The mood remains inquisitive and playful as the speaker engages directly with the audience. The clip transitions with the speaker confidently revealing,... His tone conveys satisfaction, and he describes the illusion as one of his favorites, adding that it is simple to perform. ... He jokingly acknowledges his lack of enthusiasm for fulfilling this promise, all while maintaining the lighthearted and engaging atmosphere. The clip concludes with an uplifting tone, inviting further interaction from the audience.

**Overall Audio Description:**
The audio begins with an enthusiastic and friendly voice greeting, ... The speaker, speaking with a slight accent, warmly sets the stage for the presentation while teasing a "very cool illusion" to generate curiosity. The background remains quiet as the speaker describes the visual illusion, mentioning ... and invites the audience to actively engage by asking, "What do you see?" The speaker's tone is playful and inquisitive ... The tone transitions to one of confidence and satisfaction as the speaker reveals... The informal and friendly connection persists as they conclude the presentation with, "Alright guys, that's pretty much it," ... Humor enters the tone as the speaker reluctantly references a previous promise to perform a cinnamon challenge ... The audio moves seamlessly into a closing segment, where the speaker cheerfully thanks the audience with, ' "Thank you for watching," ... The overall tone concludes on a positive, friendly note, emphasizing community and connection with the audience.

**Video Caption:**
The video begins with a person wearing a white lab coat and an orange shirt... Next, the focus shifts to a white sheet of paper with two drawn tables,... A hand points at the black-topped table, then the red-topped table, indicating a comparison or explanation. Two hands then interact with the tables, ... The video returns to the person in the white lab coat and orange shirt,... They gesture with the bottle, turning it slightly to show different angles, and then set the bottle down, continuing to speak. The background remains the same, with the various posters still visible. Throughout the video, the person maintains a focused and engaged demeanor, consistently seated in front of the visually engaging backdrop of posters, explaining or demonstrating various points.

**Generated QA:**
**Question:** How does the speaker demonstrate the optical illusion with the two tables?
**Answer:** The speaker first describes the illusion with enthusiasm, mentioning the two tables with different shapes. As he points to each

---

> table and gestures with his hands, the visual of moving the black and red rectangles is shown, aligning with his explanation that they are actually the same size despite appearing different.

We illustrate sample data in Table 3.2, emphasizing the temporal structure in audio annotations and the extraction of nuanced emotional cues from audio. Our approach effectively captures complex temporal relationships across audio segments and explicitly highlights challenging emotional aspects. The generated QA pairs specifically require integrated audio-visual reasoning, demonstrating their potential to advance multimodal understanding. Detailed prompts used for QA generation is shown in Appendix B.

**Dataset Statistics** Our dataset comprises 80k training examples and 1k benchmark QA pairs, sourced entirely from YouTube to ensure diverse and generalized video content. Videos range from 5 seconds to approximately 2 minutes (120.48 seconds), with an average duration of 37.99 seconds. Specifically, around 60k videos fall within the 0–30 second interval, while the remaining 20k span between 1–2 minutes. Audio descriptions vary from 32 to 484 tokens, averaging 131.78 tokens. Video captions contain between 53 and 1819 tokens, averaging 392.34 tokens. Detailed statistics are summarized in Table 1.

Table 1: Statistics of SoundInSights data.

|  | min | max | mean |
| --- | --- | --- | --- |
| Video Duration (seconds) | 5.00 | 120.48 | 37.99 |
| Audio Description (#tokens) | 32 | 484 | 131.78 |
| Video Caption (#tokens) | 53 | 1819 | 392.34 |

## 3.3 SOUNDINSIGHTS BENCHMARK

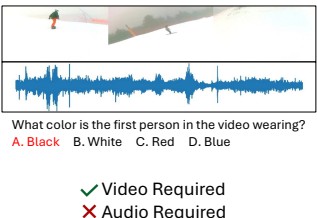 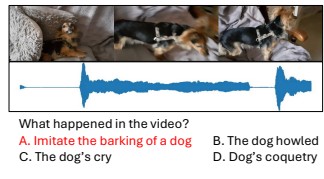 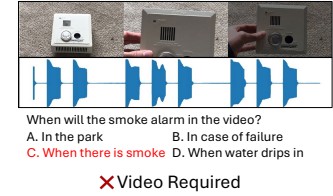

Figure 3: Example from AVQA Yang et al. (2022), where the samples are directly retrieved from their appendix as representative examples.

When constructing an audio-visual multimodal understanding benchmark, it is crucial to ensure that questions inherently require both video and audio modalities for reasoning. If a model can correctly answer questions using only a single modality (either visual or auditory), two critical issues arise: (1) The model may achieve high accuracy without genuinely establishing cross-modal associations, making the benchmark ineffective for assessing multimodal understanding capabilities; (2) it can lead to a false correctness phenomenon when one modality is missing, where the model hallucinates correct answers based on dataset biases or prior knowledge rather than true multimodal reasoning. This limitation has been observed in existing research. For example, the AVQA benchmark Yang et al. (2022), as illustrated in Figure 3, includes questions that can often be correctly answered using only visual or auditory information, or even general common sense reasoning.

Besides high-quality multimodal data for fine-tuning, a challenging evaluation benchmark is also essential for developing audio-visual multimodal large language models (MLLMs). To this end, we introduce the *SoundInSights Benchmark*, a challenging benchmark comprising around 1,000 carefully curated videos accompanied by corresponding open-ended questions. To guarantee high data quality, we adopt a human-in-the-loop approach rather than fully automated generation. Initially, approximately 4,000 candidate videos with strong audio-video correlations and substantial auditory content are identified using ChatGPT. Subsequently, we manually select a refined subset, ensuring the final 1,000 videos have clear visuals, rich and relevant auditory elements, and questions that are meaningful, objective, and sufficiently challenging. For robust and stable model evaluation, we employ a multiple-choice format rather than relying directly on an LLM-as-Judge paradigm to compare model outputs against ground truth.

## 3.4 MODEL STRUCTURE AND TRAINING STRATEGY

To incorporate the auditory modality, we adopt the widely-used LLaVA's architecture Liu et al. (2023)(see Figure 4), which encode signal from different modalities into language embeddings and *directly append* them together. Specifically, in addition to the visual encoder and projector, we introduce an *audio encoder* to process the raw audio signals into high-level audio features. These features are then passed through a *projector* to align with the existing latent space of the LLM.

To facilitate the training process, we adopt the audio encoder of Qwen2Audio Chu et al. (2024) for our model, which has been pretrained on large-scale audio data.

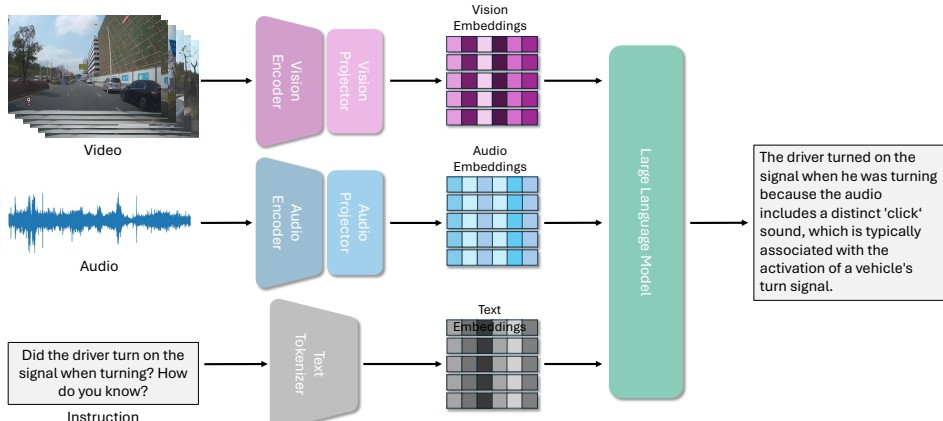

Figure 4: Model Structure: We follow the classic LLaVA design and add an audio branch that projects acoustic tokens into the LLM latent space.

**Model Training.** Instead of training from a text-only LLM base model, we further fine-tune a well-pretrained multi-modal large language model, LLaVA-One-Vision-7B Li et al. (2024), which is capable of performing video understanding but cannot perceive audio content. This strategy can alleviate a lot of training resources.

However, during training, we find that model directly fine-tuned on the question-answering pairs in SoundInSights will hallucinate audio contents given visual signals, which indicates that the model only learns the superficial "format" of audio reasoning instead of trying to understand the audio contents.

A plausible reason is that, these pre-trained model has built strong connections between vision and language, which makes it hard to inject audio information from an un-aligned audio representation.

To this end, we devise a two-stage training strategy which first align the audio tower with the pre-trained vision-language model.

In the first stage, we completely remove the vision encoder and inputs, and train the audio projector on 80k caption data in the proposed SoundInSights dataset. The training task is then degraded into audio captioning. Without the help of visual signals, the vision-language model is forced to extract information from the audio representation and thereby achieves better alignment.

The second stage aims to enable the model to jointly consider both visual and audio input. We collect 80k caption data with 80k audio-video question answering data from the SoundInSights , combined with 80k question-answering data sampled from LLaVA-Video-178K to preserve model's ability on general visual question answering. At this stage, both the vision projector and the audio projector is updated to allow the representations of all modalities to co-adapt, yielding more robust alignment and better multimodal understanding.

Table 2: Performance of LLaVA-OneVision-Qwen2 7B model and ours. As shown in the table, with SFT on SoundInSights data, our model achieves significant improvement on temporal perception and information synopsis tasks.

| | VideoMME | | | | | | | SoundInSights |
| | Overall | Short | Medium | Long | Temp. Pcpt. | Temp. Rsn. | Info. Synopsis | Overall |
|---|---|---|---|---|---|---|---|---|
| LLaVA OV 7B | 57.44 | 69.8 | 55.4 | 47.1 | 56.4 | **43.5** | 67.8 | 85.6 |
| Ours | **58.11** | **71.1** | **55.7** | **47.6** | **63.6** | 40.1 | **75.9** | **92.7** |

Table 3: Performance of our base model and ablation study without audio caption pretraining.

| | VideoMME | | | | | | | | | SoundInSights |
| | overall | Short | Medium | Long | Temp. Pcpt. | Temp. Rsn. | Spc. Pcpt. | Spc. Rsn. | Info. Synopsis | Overall |
|---|---|---|---|---|---|---|---|---|---|---|
| Ours | 58.11 | 71.1 | 55.7 | 47.6 | 63.6 | 40.1 | 61.1 | 82.1 | 75.9 | 92.7 |
| w/o pretrain | 56.0 | 68.3 | 54.1 | 45.7 | 60.0 | 37.3 | 57.4 | 78.6 | 70.9 | 90.7 |

## 4 EXPERIMENTS

### 4.1 IMPLEMENTATION DETAILS

We choose LLaVA-OneVision Qwen2 7B Li et al. (2024) as our base model. We integrate the audio encoder from Qwen2-Audio 7B Chu et al. (2024) together with a two-layer MLP with GELU activation Hendrycks & Gimpel (2016) as our audio tower.

**Training details** At the *Stage 1* training, we train the audio projector with 80k level-3 audio captions from SoundInSights dataset. The learning rate is $2e - 6$ and the batch size is 128 as 4 samples on each of 8 Nvidia A100 GPUs with gardient accumulation steps of 4. The stage 1 training runs 2 epochs on the SoundInSights dataset. At the *Stage 2* training, we unfreeze both the video projector and the audio projector. To avoid overfitting on audio video joint QA scenarios, we mix the 80k open-ended audio video QAs with randomly sampled 100k open-ended QAs and 100k multi-choice questions from the LLaVA-178K dataset Zhang et al. (2024). For each of questions from this training set, both video and audio are fed into the model, thus the model can capture all the information to answer any type of the question. The learning rate is $1e - 5$ and batch size is 32 as 1 sample on each of 8 Nvidia A100 GPUs with gradient accumulation steps of 4.

**Benchmarks** We evaluate our model on two benchmark: VideoMME Fu et al. (2024) and our SoundInSights benchmark. VideoMME is a general video understanding benchmarks designed for MLLM, which mostly consists of vision-centric question-answering tasks. However, it also has audio track for the video. Hence, we test our model with the audio input on VideoMME.

### 4.2 RESULTS

**Qualitative Examples** We show some examples of our model with the baseline model LLaVA-OneVision-7B in Figure 5, where both video and audio contents should be taken into consideration. For the first case, the man slips while kicking the ball, and splat the drink in his hand. It can be hard that the other man says that the football is attached with a string which makes the man slips. Our model successfully captures this information and gives the correct answer. However, the baseline model determine that the spilled drink from the man's fall caused him to slip, which reverses cause and effect. For the second case, the speaker is completely invisible during the video. While her voice of encouragement can be clearly captured by our model.

**Quantitative Results** The experimental results presented in Table 2 clearly illustrate that our model significantly outperforms LLaVA-OneVision-Qwen 7B model across multiple evaluation metrics, highlighting the effectiveness of our tailored dataset and training approach. Notably, the introduction of our SoundingSights dataset, which explicitly emphasizes audio-visual joint understanding, has markedly improved performance on tasks requiring integrated modal reasoning. Our model's substantial gains demonstrate that conventional multimodal models struggle with nuanced audio-video interactions due to their more general training data. Conversely, our dataset's specific design to facilitate joint modal comprehension enables our model to excel, particularly in scenarios demanding combined audio and visual inference. This suggests that targeted, modality-integrative datasets like SoundingSights are essential for advancing multimodal reasoning capabilities.

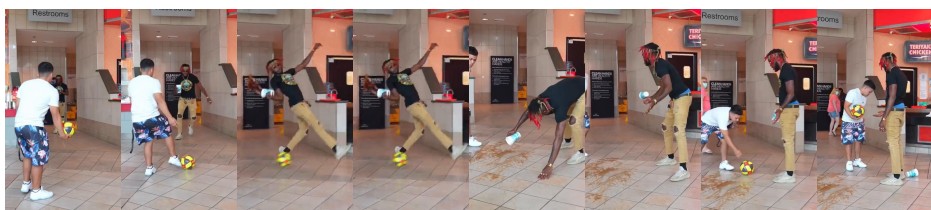

**Input Question:**
*Why does the second man slip and spill his drink when approaching the soccer ball in the hallway?*
*A. He wore oversized shoes that made him trip.*
*B. The floor was already wet from a previous spill.*
*C. The soccer ball was attached to a hidden string.*
*D. He slipped on a loose tile by the hallway.*

**Model Output:**
*LLaVA-OV-7B: B.*
*Ours: C.*

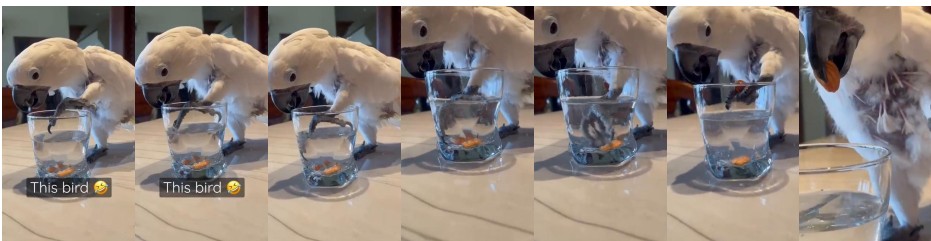

**Input Question:**
*While the parrot tries to retrieve an orange object from the glass, how does the person's voice line up with its progress?*
*A. They keep telling jokes, causing the parrot to peck uncertainly all along.*
*B. They stay completely silent until the parrot fully abandons its attempt.*
*C. They encourage the parrot during its struggles and cheer upon success.*
*D. They issue random commands that confuse the parrot and block any progress.*

**Model Output:**
*LLaVA-OV-7B: A.*
*Ours: C.*

Figure 5: Qualitative Results: We visualize our results together with LLaVA-OV-7B results. As shown in the figure, our models successfully captures both of visual and auditory information.

### 4.3 ABLATION STUDY

We conduct ablation study on the effect of pretraining on audio caption task. Pretraining on audio-caption data significantly boosts performance across all metrics by teaching the model richer multimodal representations before fine-tuning. As seen in Table 3, the pretrained model consistently surpasses its non-pretrained counterpart, especially on tasks involving temporal, spatial, and holistic understanding. This highlights how aligning audio signals equips the model with a more robust feature space, enabling better perception, reasoning, and synopsis capabilities in diverse video settings.

## 5 CONCLUSION

We presented a new multimodal dataset for video understanding that requires joint reasoning over both visual and auditory streams, accompanied by a scalable MLLM-driven labeling pipeline. By providing hierarchical audio annotations and audio-visual QA pairs, our dataset offers a challenging benchmark which goes beyond existing visual-centric tasks. We also introduced a baseline model demonstrating the benefits of integrating sound cues for improved comprehension.

**Limitations and Future Work.** This work aims to prove the effectiveness of multimodal data annotated by MLLMs for video understanding. We fuse visual features with audio features by simple concatenation. We leave the exploration of more efficient way of fusing different modalities for future work.

**Broader Impacts.** Enhanced audio-visual systems promise better assistive captions, safer embodied agents, and richer retrieval. We hope that the proposed dataset, benchmark, and pipeline will catalyze further research in holistic audio-visual intelligence and lead to more robust, context-aware video understanding systems.

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
