# A    PROMPTS FOR AUDIO ANNOTATION

Table 4: **Prompt templates for hierarchical audio captioning**: The first clip uses `first_clip_system`, subsequent clips use `other_clip_system`. Every $n = 2$ clips we invoke `summary_system` to obtain a running narrative, and once all clips are processed we request a final video-level audio description.

---

**Variables**
`first_clip_system` = "### Task:
You are an expert in understanding scenes based on audio features in a video ..."
`first_clip` = "Please provide a detailed description of the ..."
`other_clip_system` = "### Task:
You are an expert audio analyst specializing in ..."
`other_clip` = "Please provide a detailed description of the current audio clip ..."
`summary_system` = "### Task:
You are an expert at creating comprehensive audio descriptions ..."
`summary_user` = "### Clip Descriptions (chronological):
{all_clip_caption}
Please give me the summary description ..."

**Generation Loop**
```
for audio in audios:
    clips = split_into_30s(audio)
    for idx, clip in enumerate(clips):
        if idx == 0 then
            sys_msg = first_clip_system
            usr_msg = first_clip.format(video_caption)
        else
            sys_msg = other_clip_system
            usr_msg = other_clip.format(previous_captions,
            video_caption)
        clip_caption = GPT-4o_Audio(sys_msg, usr_msg, clip)
        captions.append(clip_caption)
        if need_summary then
            sys_msg = summary_system.format(#clips, rand_structure)
            usr_msg = summary_user.format(all_clip_caption=captions)
            summary = GPT-4o(sys_msg, usr_msg)
            summaries.append(summary)
```

**Output Format**
Each assistant reply must be a JSON dictionary with a single key:
`"Clip Level Caption"` *or* `"Video Level Audio Description"` as required.

---

# B    PROMPTS FOR QA GENERATION

Table 5: **Prompt for generating multimodal QA pairs**: For each video, the video-level caption and its hierarchical audio captions are supplied to GPT-O1. The model is instructed to output a single question–answer pair whose answer demands joint temporal and semantic reasoning over both visual and auditory cues.

```
captions = "
### Video description: {video_caption}
### Audio clip captions: {clip_caption}
### Audio summary description: {audio_caption}

### Task:
Generate 1 QA pair that REQUIRES analyzing BOTH:
1. Visual elements
2. Audio elements
Focus on their TEMPORAL and SEMANTIC relationships. "
system_message = "
### Task: You are an expert in analyzing multimodal content. Based on the provided descriptions of
a scene's visuals and sounds, generate a single question and answer pair that REQUIRES combining
information from BOTH the video (visual) and the audio (sound events). The question must hinge on
details from both modalities.
### Input: 1. Visual summary (describing the scene, objects, or actions).
2. Audio summary (listing or describing the key sounds, their timing, and context).

### Guidelines:
1. Focus on how the audio and visual elements INTERACT:
- How specific sounds correlate with visible actions or objects
- How audio provides context missing from visuals (or vice versa)
- Their temporal alignment (e.g., a sound occurring exactly when something is seen)

2. Strict requirements:
- Do NOT ask about purely audio-only or purely video-only details
- Do NOT explicitly mention "audio description" or "video description" in your question or answer.
- If there is NO meaningful audio-visual interplay (i.e., the audio doesn't add unique info to the
visuals or vice versa), return:
{"Question": "NA", "Answer": "NA"}

3. Format:
- Return your output as a **JSON object** in a Python dictionary string format.
- It must have the keys "Question" and "Answer" only.

### Example: Here's a sample output for a dash-cam video showing a car turning with an audible
turn-signal click:
{"Question": "Does the driver activate the turn signal before turning?", "Answer": "Yes, the blinking
light is visible and the distinct clicking sound is heard at the moment of the turn."} "

for cur_video in videos:
    sys_msg = system_message
    usr_msg = captions.format(
    video_caption=cur_video.video_caption,
    clip_caption=cur_video.clip_caption,
    audio_caption=cur_video.audio_caption)
    response = GPT-o1(sys_msg,usr_msg)
```