# OpenReview forum: "Sound in Sights: Deriving Visual Insights from Audio for Comprehensive Video Understanding with Large Multimodal Models"
_ICLR.cc/2026/Conference — ICLR 2026 Conference Withdrawn Submission_

### Official Review · Reviewer_BCn2 · 2025-10-19

**Soundness:** 3
**Presentation:** 2
**Contribution:** 3
**Rating:** 4
**Confidence:** 4

**Summary:**

This paper introduces SoundInSights, a large-scale (approx. 80k QA pairs) audio-visual corpus created with hierarchical audio annotations to encourage genuine joint AV reasoning. An additional curated benchmark with 1,000 QA pairs is also proposed to assess tasks requiring audio-visual integration. The authors enhance a vision-centric MLLM (LLaVA-OneVision) by adding an audio encoder (Qwen2-Audio encoder) and propose a two-stage training method: audio-only caption pre-training followed by joint AV fine-tuning. Experiments show clear improvements on their benchmark and VideoMME evaluation, confirming the value of their approach.

**Strengths:**

The authors clarified the problem motivation, many events are “visible in sound” but not in frames. Figure 1 illustrates the "turn signal" case where audio modality prevents hallucinations from pure visual input. A simple and reproducible training recipe is also proposed, the two-stage tuning scheme is straightforward and effective, with implementation details (e.g., learning rate, batch, computational resource usage) are given. Apart from these, well-designed supervision signals are also presented, a hierarchical audio annotating mechanism is proposed to encourage the modeling of temporal structure and both fine- and coarse-grained audio semantics.

Results suggest that models over rely on visual modality, with both quantitative improvements on audio related tasks (Tab. 2) and qualitative cases support the motivation (Figure 5).

**Weaknesses:**

1. Insufficient Baseline Comparisons

The paper primarily compares against a baseline model (LLaVA-OV-7B) that lacks audio processing capabilities. This restricts the conclusions that can be drawn regarding the effectiveness of the proposed method. To convincingly demonstrate broader applicability, comparisons against current state-of-the-art audio-visual models such as video-SALMONN, VideoLLaMA2, AVicuna, or Macaw-LLM are necessary (Even suggesting that the proposed approach narrows the gap is also meaningful).

2. Unexplained Performance Drop in Specific Tasks

On the VideoMME benchmark, the Temporal Reasoning subtask performance notably decreases from 43.5 to 40.1. The authors should investigate and explain this unexpected decline, potentially analyzing alignment strategies, handling of temporal dependencies, or error patterns to provide clarity.

3. Potential Issues with Data Overlap and Leakage

Since the dataset is derived from videos included in LLaVA-Video-178K, there is a possibility of overlap or unintended leakage between the training and test data. The paper currently does not detail explicit procedures for data deduplication at the video, audio, or text level, raising concerns regarding the integrity of the evaluation setup.

4. Limited Evaluation Breadth

The evaluation is restricted to the newly proposed benchmark and VideoMME. This lack of testing on established audio-visual benchmarks such as AVQA, AVSD, MUSIC-AVQA, AV-Odyssey, or AV-TrustBench limits the generalizability and external validity of the findings. Further experimentation on these additional tasks would strengthen confidence in the proposed approach.

Minor issues

1. There are discrepancies in reporting the dataset sizes used in Stage 2 training (e.g., inconsistently described as "80k AV-QA plus 80k LLaVA-Video QAs" in Sec 3.4 versus "80k AV-QA plus 100k open-ended plus 100k multi-choice" in Sec. 4.1).
2. Minor typos such as "gardient" and "SoundingSights".

**Questions:**

My questions primarily lie in the following perspectives:

## Dataset construction
1. Data leakage prevention
How did you ensure no overlap between the SoundInSights 80k training set and the 1k benchmark, and between your training data and VideoMME clips? Especially since sources are from YouTube via LLaVA‑Video‑178K.

2. Validation of "both modalities are required"
For the proposed 1k multiple‑choice benchmark, please report the A‑only / V‑only / A+V accuracies to demonstrate that questions truly require cross‑modal reasoning?

3. Quality control
The hierarchical audio descriptions use GPT‑4o and the 1k benchmark was curated from the approximately 4k ChatGPT filtered candidates. Was there any quality control process performed?

4. Benchmark Composition and Difficulty
Please share the topic/source distribution, typical temporal spans, and answer distribution of the 1k benchmark. This would further confirm coverage beyond a few easy tropes and calibrate multiple choice chance levels.

## Method
1. Inconsistent Stage‑2 mixture sizes
Sec. 3.4 says Stage‑2 mixes 80k AV‑QA + 80k LLaVA‑Video QAs, while Sec. 4.1 reports 80k AV‑QA + 100k open‑ended plus 100k multi‑choice. Which one is correct?

2. Stage‑1 supervision granularity
As stated in the manuscript, Stage‑1 trains on 80k Level‑3 audio captions. Did you evaluated the performance of using Level‑1/Level‑2 as well, or curriculum from fine‑ to coarse‑grained?

3. Feature fusion
The architecture concatenates audio and vision embeddings into the LLM. Did you evaluate cross‑attention or gated fusion to mitigate modality interference?

4. Sampling rate
What are the frame sampling rate, clip length, audio sampling rate, and token counts per modality during inference?

## Evaluation protocol and baseline fairness
1. Baseline models
Please also include comparisons with video-SALMONN, VideoLLaMA2, AVicuna, Macaw‑LLM... under the same evaluation protocol.

2. A‑only / V‑only / A+V ablations on external benchmarks
On VideoMME, please also report performance when feeding only video, only audio, and both to your model.

3. Broader benchmark coverage
Please also report results on datasets such as AVQA, AVSD, MUSIC‑AVQA, AV‑Odyssey, AV‑TrustBench (maybe not all of them). This would demonstrate generalization beyond your own benchmark and one general purpose set.

## Analysis of robustness and failure modes
1. Why does Temporal Reasoning decrease (from 43.5 to 40.1)? Is the audio branch over‑attended or mis‑aligned with the frame timing?

2. What happens if you time‑shift the audio, swap with other clips, or mute segments? This would demonstrate whether the model uses audio rather than the priors.

## Minor issues

Please also fix the "SoundingSights" and typos like “gardient”.

---

### Official Review · Reviewer_a33s · 2025-10-28

**Soundness:** 1
**Presentation:** 1
**Contribution:** 1
**Rating:** 2
**Confidence:** 5

**Summary:**

This paper introduces SoundInSights, an audio-visual question answering dataset based on LLaVA-Video-178k.

**Strengths:**

- This work extends the annotations of LLaVA-Video-178k by adding annotations for audio information.

**Weaknesses:**

- There are already well-annotated video datasets that consider audio, such as Finevideo and Cinepile. If annotation based on MLLM is intended, models with SOTA performance on audio-visual data can also be used for annotation, such as Gemini-2.5, video-SALMONN 2, and Qwen2.5-Omni. The annotation method used in this paper separates audio descriptions from visual descriptions, ignoring the synchronization of audio and video.
- The annotations of SoundInSights are completely generated by LLMs, lacking manual verification.
- SoundInSights is entirely based on LLaVA-Video-178k and only serves as an extension of it, resulting in very limited novelty.
- In Table 2, the improvement on VideoMME is actually very limited, and it might even be caused by errors. Moreover, the tested benchmarks are too few. Only VideoMME is tested among public benchmarks. All these lead me to question the effectiveness of the SoundInSights dataset.

**Questions:**

- See Weaknesses

---

### Official Review · Reviewer_KKKJ · 2025-10-30

**Soundness:** 2
**Presentation:** 3
**Contribution:** 2
**Rating:** 4
**Confidence:** 5

**Summary:**

This paper aims to solve the problem wherein most existing Multimodal Large Language Models (MLLMs) are restricted to video understanding and cannot receive audio input. A scalable MLLM-assisted annotation pipeline for generating hierarchical audio descriptions and context-aware AV QA pairs is proposed. Based on this, the SOUNDINSIGHTS training set, containing 80k samples, and a human-verified test set are introduced. Using LLaVA-One-Vision-7B as the base, the authors adopted a 2-stage training method, confirming the importance of audio information for MLLMs.

**Strengths:**

- Exploring end-to-end audio understanding for MLLMs is a promising research direction.
- The manuscript is clearly written and easy to follow.
- The promise to open-source the data and code is a valuable contribution to the paper's reproducibility.

**Weaknesses:**

Honestly, I find that this paper, in its current form, does not yet meet the standards for publication.
- The paper's novelty is insufficient. The fine-grained use of MLLMs to assist in constructing training data has already been proposed in prior work such as [1]. Furthermore, the model's architectural design is outdated.
- The experimental validation is inadequate. Firstly, the method is not compared against a sufficient number of strong baselines, particularly recent SOTA models like video-SALMONN-2+ and Qwen2.5/3-Omni. Secondly, the evaluation is conducted on a limited selection of benchmarks. To substantiate the paper's claims, the analysis must be broadened to include other established benchmarks like WorldSense, AVUT, and Video-Holmes, in addition to VideoMME.
- There is a clear gap between the model's performance and the state-of-the-art. Many 7B models, for instance, achieve scores over 60 on VideoMME. The reported performance is substantially lower, which makes it difficult to be convinced of the paper's contributions.


[1] Sun et al., "video-SALMONN-o1: Reasoning-enhanced Audio-visual Large Language Model", ICML 2025.

**Questions:**

Please refer to Weaknesses.

---

### Official Review · Reviewer_Uv3i · 2025-10-30

**Soundness:** 2
**Presentation:** 2
**Contribution:** 2
**Rating:** 4
**Confidence:** 5

**Summary:**

The authors present a high-quality audio-visual question answering dataset, a Benchmark, along with its accompanying construction workflow. When applied to the fine-tuning phase of AVLLMs, this dataset and workflow can enhance general video understanding Benchmarks, such as VideoMME.

**Strengths:**

1. The authors provide a high-quality dataset and Benchmark for joint audio-visual understanding, which contributes to the further development of the field.
2. By using the provided dataset to train audio-visual understanding models and achieving performance improvements on VideoMME, the authors have verified that the dataset is beneficial for audio-visual understanding.

**Weaknesses:**

1. Although the dataset and Benchmark provided by the authors hold certain value for the field, they fail to elaborate on the essential differences between the proposed dataset and Benchmark. The work can only be regarded as a supplement to previous studies, thus making a relatively small contribution to advancing the field of audio-visual understanding.
2. When constructing high-quality audio-visual data, the key lies in creating QAs (Question-Answers) or Captions that require simultaneous use of audio and visual information for responses. While the authors have attempted to propose some solutions, these solutions have not been properly validated or explained.
3. Multi-granularity synthetic data has long been proposed in LLaVA-Video [1] and widely adopted in many subsequent works, so it cannot be considered an original contribution. Furthermore, from the perspectives of model design and training strategies, the authors have not put forward any key technical contributions either.

[1] LLaVA-Video: Video Instruction Tuning With Synthetic Data

**Questions:**

1. What is the essential difference between SoundInSights and previous Benchmarks and datasets, especially more advanced ones such as AV-Odysee [1]?
2. Why can the authors' data construction scheme synthesize content that mandates reasoning based on both audio and visual information? The authors need to verify and elaborate on this question.

[1] AV-Odyssey Bench: Can Your Multimodal LLMs Really Understand Audio-Visual Information?

---

### Note · Authors · 2025-11-12

I have read and agree with the venue's withdrawal policy on behalf of myself and my co-authors.